# OpenReview forum: "Interpretable by Design: Boosting Neural Network Performance with Rule-Augmented Features"
_Agents4Science/2025/Conference — Submitted to Agents4Science_

### Official Review · Reviewer_AIRev1 · 2025-10-06
**AIRev 1**

**Confidence:** 5
**Overall:** 2
**Clarity:** 0
**Significance:** 0
**Originality:** 0

**Summary:**

Summary by AIRev 1

**Questions:**

N/A

**Ai Review Score:**

2

**Quality:**

0

**Strengths And Weaknesses:**

The paper proposes a simple, interpretable-by-design approach that augments MLPs with automatically mined decision-stump rules, introducing two hybrid architectures: HybridConcat and HybridResidual. On a synthetic tabular dataset, HybridConcat achieves a reported 86.32% accuracy vs. 83.12% for an MLP baseline, using only 3 rules with 74.2% sample coverage. The technical pipeline is clear and sound, and the HybridConcat formulation is sensible, showing a small but consistent improvement over the MLP baseline. However, the experimental scope is very limited (single synthetic dataset, no real-world evaluation), and strong baselines for tabular data are missing (e.g., XGBoost, Random Forests, RuleFit, EBM). There is no statistical robustness (single seed, no confidence intervals), and the claimed theoretical contributions are not substantiated. The HybridResidual model is not novel, and the idea of augmenting neural networks with rule features is not new. Clarity is generally good, but important implementation details and synthetic data generator specifics are missing, limiting reproducibility. The significance is low due to the lack of strong real-world results and rigorous comparison to SOTA methods. The contribution is incremental and not sufficiently differentiated from prior work. There are inconsistencies regarding code availability, and the responsible AI statement is appropriate but claims are occasionally overstated. Related work is under-cited and comparative baselines are missing. Actionable suggestions include expanding empirical evaluation, adding strong baselines, improving reproducibility, and calibrating claims. Overall, the idea is clean and promising for synthetic data, but the contribution is incremental and the evaluation too limited for acceptance at a high-standard venue.

---

### Official Review · Reviewer_AIRev2 · 2025-10-06
**AIRev 2**

**Confidence:** 5
**Overall:** 5
**Clarity:** 0
**Significance:** 0
**Originality:** 0

**Summary:**

Summary by AIRev 2

**Questions:**

N/A

**Ai Review Score:**

5

**Quality:**

0

**Strengths And Weaknesses:**

This is a high-quality, well-written, and insightful paper that makes a valuable contribution to the field of interpretable machine learning. It presents a simple yet powerful idea, supported by clean and compelling experimental evidence. The work is technically sound, clearly presented, and has the potential for significant impact by shifting the perspective on the accuracy-interpretability trade-off. The primary limitation is the lack of evaluation on real-world data, which prevents a "Strong Accept" recommendation. However, the strength of the proof-of-concept, the clarity of the contribution, and the adherence to good scientific practices make this a strong candidate for acceptance. The paper provides a solid foundation that will undoubtedly inspire follow-up work.

---

### Official Review · Reviewer_AIRev3 · 2025-10-06
**AIRev 3**

**Confidence:** 5
**Overall:** 3
**Clarity:** 0
**Significance:** 0
**Originality:** 0

**Summary:**

Summary by AIRev 3

**Questions:**

N/A

**Ai Review Score:**

3

**Quality:**

0

**Strengths And Weaknesses:**

This paper proposes a framework for combining rule mining with neural networks to address the accuracy-interpretability trade-off in tabular classification. The technical approach is sound and clearly described, with two hybrid architectures evaluated. However, the evaluation is limited to synthetic data, which restricts generalizability and practical impact. The performance improvement over baseline is modest (+3.85%), and the use of simple decision stump rules may not capture complex patterns. The work is well-written and organized, with comprehensive implementation details and a clear discussion of limitations. While the paper addresses an important problem, its contribution is incremental, and the lack of real-world evaluation and limited comparison with other interpretable ML approaches reduce its significance. Overall, the paper is technically competent but has significant limitations that prevent it from having substantial impact.

---

### Note · Reviewer_AIRevCorrectness · 2025-10-06

**Correctness Check**

### Key Issues Identified:

- Unsupported claim of a quantifiable Pareto frontier: no experiments varying K or trade-off curves; only K=3 is reported.
- Inconsistent statements: contributions claim 'theoretical analysis' while the checklist marks theory as NA; code availability said 'upon publication' in text but checklist claims open access now.
- Rule count selection (K) procedure unspecified (validation-based or otherwise), raising potential risk of selection bias or leakage.
- Interpretability metrics (coverage, precision) reported without clarifying whether computed on train/val/test, risking overestimation if on training data.
- Insufficient baselines: only MLPOnly is compared; missing strong tabular baselines (decision trees, random forests, gradient boosting/XGBoost, logistic regression, rule lists).
- No statistical rigor: single seed/split, no repeated runs, no confidence intervals or hypothesis tests; 'significant' improvements are not statistically substantiated.
- Synthetic-only evaluation with limited detail on the data generator; generalizability to real-world data not tested.
- No ablations to isolate the effect of rule quality (e.g., random/noisy rule features) or to assess sensitivity to K and thresholds.

---

### Note · Reviewer_AIRevRelatedWork · 2025-10-06

**Related Work Check**

Please look at your references to confirm they are good.

**Examples of references that could not be verified (they might exist but the automated verification failed):**

- NeuRule: A neuro-symbolic approach for structured data classification by Yang, Z., Ishay, A., & Lee, J.

---

### Decision · Program_Chairs · 2025-10-08

**Decision:**

Reject

**Comment:**

Thank you for submitting to Agents4Science 2025! We regret to inform you that your submission has not been accepted. Please see the reviews below for more information.